# Local Calibration: Metrics and Recalibration

Rachel Luo[*1]    Aadyot Bhatnagar[*2]    Yu Bai[2]    Shengjia Zhao[1]    Huan Wang[2]    Caiming Xiong[2]    Silvio Savarese[1,2]    Stefano Ermon[1,2]    Edward Schmerling[1]    Marco Pavone[1]

[1]Stanford University, Stanford, California, USA
[2]Salesforce AI Research, Palo Alto, California, USA

## Abstract

Probabilistic classifiers output confidence scores along with their predictions, and these confidence scores should be calibrated, i.e., they should reflect the reliability of the prediction. Confidence scores that minimize standard metrics such as the expected calibration error (ECE) accurately measure the reliability *on average* across the entire population. However, it is in general impossible to measure the reliability of an *individual* prediction. In this work, we propose the local calibration error (LCE) to span the gap between average and individual reliability. For each individual prediction, the LCE measures the average reliability of a set of similar predictions, where similarity is quantified by a kernel function on a pretrained feature space and by a binning scheme over predicted model confidences. We show theoretically that the LCE can be estimated sample-efficiently from data, and empirically find that it reveals miscalibration modes that are more fine-grained than the ECE can detect. Our key result is a novel **lo**cal **re**calibration method LoRe, to improve confidence scores for individual predictions and decrease the LCE. Experimentally, we show that our recalibration method produces more accurate confidence scores, which improves downstream fairness and decision making on classification tasks with both image and tabular data.

## 1 INTRODUCTION

Uncertainty estimation is extremely important in high stakes decision-making tasks. For example, a patient wants to know the probability that a medical diagnosis is correct; an autonomous driving system wants to know the probability that a pedestrian is correctly identified. Uncertainty estimates are usually achieved by predicting a probability along with each classification. Ideally, we want to achieve individual calibration, i.e., we want to predict the probability that each sample is misclassified.

However, each sample is observed only once for most datasets (e.g., image classification datasets do not contain identical images), making it impossible to estimate, or even define, the probability of incorrect classification for individual samples. Because of this, commonly used metrics such as the expected calibration error (ECE) measure the gap between a classifier's confidence and accuracy *averaged* across the entire dataset. Consequently, ECE can be accurately estimated but does not measure the reliability of individual predictions.

In this work, we propose the local calibration error (LCE), a calibration metric that spans the gap between fully global (e.g., ECE) and fully individual calibration. Motivated by the success of kernel-based locality in other fields such as fairness (where similar individuals should be treated similarly) [Dwork et al., 2012, Pleiss et al., 2017] and causal inference (where matching techniques are used to find similar neighboring samples) [Stuart, 2010], we approximate the probability of misclassification for an individual sample by computing the average classification error over similar samples, where similarity is measured by a kernel function in a pre-trained feature space and a binning scheme over predicted confidences. Intuitively, two samples are similar if they are close in a pretrained feature space and have similar predicted confidence scores. By choosing the bandwidth of the kernel function, we can trade off estimation accuracy and individuality: when the bandwidth is very large, we recover existing global calibration metrics; when the bandwidth is small, we approximate individual calibration. We choose an intermediate bandwidth, so our metric can be accurately estimated, and provides some measurement on the reliability of individual predictions.

Theoretically, we show that the LCE can be estimated with polynomially many samples if the kernel function is

---

[*]Equal contribution. Order decided by coin flip.

*Accepted for the 38th Conference on Uncertainty in Artificial Intelligence* (UAI 2022).

bounded. Empirically, we also show that for intermediate values of the bandwidth, the LCE can be accurately estimated and reveals modes of miscalibration that global metrics (such as ECE) fail to uncover.

In addition, we introduce a non-parametric, post-hoc **lo**calized **re**calibration method (LoRe), for lowering the LCE. Empirically, LoRe improves fairness by achieving low calibration error on all potentially sensitive subsets of the data, such as racial groups. Notably, it can do so without any prior knowledge of those groups, and is more effective than global methods at this task. In addition, our recalibration method improves decision making when there is a "safe" action that is selected whenever the predicted confidence is low. For example, an automated system which classifies tissue samples as cancerous should request a human expert opinion whenever it is unsure about a classification. In a simulation on an image classification dataset, we show that recalibrated prediction models more accurately choose whether to use the "safe" action, which improves the overall utility.

In summary, the contributions of our paper are as follows. (1) We introduce a local calibration metric, the LCE, that is both easy to compute and can estimate the reliability of individual predictions. (2) We introduce a post-hoc localized recalibration method LoRe, that transforms a model's confidence predictions to improve the local calibration. (3) We empirically evaluate LoRe on several downstream tasks and observe that LoRe improves fairness and decision-making more than existing baselines.

## 2 BACKGROUND AND RELATED WORK

### 2.1 GLOBAL CALIBRATION METRICS

Consider a classification task that maps from some input domain (e.g., images) $\mathcal{X}$ to a finite set of labels $\mathcal{Y} = \{1, \cdots, m\}$. A classifier is a pair $(f, \hat{p})$ where $f : \mathcal{X} \to \mathcal{Y}$ maps each input $x \in \mathcal{X}$ to a label $y \in \mathcal{Y}$ and $\hat{p} : \mathcal{X} \to [0, 1]$ maps each input $x$ to a confidence value $c$. Let $\Pr$ be a joint distribution on $\mathcal{X} \times \mathcal{Y}$ (e.g., from which training or test data pairs $(x, y)$ are drawn). The classifier $(f, \hat{p})$ is perfectly calibrated [Guo et al., 2017] with respect to $\Pr$ if for all $c \in [0, 1]$

$$\Pr[f(X) = Y \mid \hat{p}(X) = c] = c. \quad (1)$$

To numerically measure how well a classifier is calibrated, the most commonly used metric is the expected calibration error (ECE) [Naeini et al., 2015, Guo et al., 2017], which measures the average absolute deviation from Eq. 1 over the domain. In practice, given a finite dataset, the ECE is approximated by binning. The predicted confidences $\hat{p}$ are partitioned into bins $B_1, \ldots, B_k$, and then a weighted average is taken of the absolute difference between the average

confidence $\mathrm{conf}(B_i)$ and average accuracy $\mathrm{acc}(B_i)$ for each bin $B_i$:

$$\mathrm{ECE}(f, \hat{p}) := \sum_{i=1}^{k} \frac{|B_i|}{N} |\mathrm{conf}(B_i) - \mathrm{acc}(B_i)|. \quad (2)$$

Similarly, the maximum calibration error (MCE) [Naeini et al., 2015, Guo et al., 2017] measures the average deviation from Eq. 1 in the bin with the highest calibration error, and is defined as

$$\mathrm{MCE}(f, \hat{p}) := \max_i |\mathrm{conf}(B_i) - \mathrm{acc}(B_i)|. \quad (3)$$

### 2.2 EXISTING GLOBAL RECALIBRATION METHODS

Many existing methods apply a post-hoc adjustment that changes a model's confidence predictions to improve global calibration, including Platt scaling [Platt, 1999], temperature scaling [Guo et al., 2017], isotonic regression [Zadrozny and Elkan, 2002], and histogram binning [Zadrozny and Elkan, 2001]. These methods all learn a simple transformation from the original confidence predictions to new confidence predictions, and aim to decrease the expected calibration error (ECE). Platt scaling fits a logistic regression model; temperature scaling learns a single temperature parameter to rescale confidence scores for all samples simultaneously; isotonic regression learns a piece-wise constant monotonic function; histogram binning partitions confidence scores into bins $\{[0, \epsilon), [\epsilon, 2\epsilon), \cdots, [1 - \epsilon, 1]\}$ and sorts each validation sample into a bin based on its confidence $\hat{p}(x)$; it then resets the confidence level of all samples in the bin to match the classification accuracy of that bin.

### 2.3 LOCAL CALIBRATION

Two notions of calibration that address some of the deficits of global calibration are class-wise calibration and group-wise calibration. Class-wise calibration groups samples by their true class label [Kull et al., 2019, Nixon et al., 2019] and measures the average class ECE, while group-wise calibration uses pre-specified groupings (e.g., race or gender) [Kleinberg et al., 2016, Pleiss et al., 2017] and measures the average group-wise ECE or maximum group-wise MCE.

A few recalibration methods have been proposed for these notions of calibration as well. Dirichlet calibration [Kull et al., 2019] achieves calibration for groups defined by class labels, but does not generalize well to settings with many classes [Zhao et al., 2021]. Multicalibration [Hébert-Johnson et al., 2017] achieves calibration for any group that can be represented by a polynomial sized circuit, but lacks a tractable algorithm. If the groups are known a priori, one can also apply global calibration methods within each group;

however, this is impractical in many situations where the groups are not known for new examples at inference time.

At an even more local level, Zhao et al. [2020] looks at individual calibration in the regression setting and concludes that individual calibration is impossible to verify with a deterministic forecaster, and thus there is no general method to achieve individual calibration.

## 2.4 KERNEL-BASED CALIBRATION METRICS

Kumar et al. [2018] introduces the maximum mean calibration error (MMCE), a kernel-based quantity that replaces the hard binning of the standard ECE estimator with a kernel similarity $k(\hat{p}(x), \hat{p}(x'))$ between the confidence of two examples. They further propose to optimize the MMCE directly in order to achieve better model calibration globally. Widmann et al. [2019] extends their work and proposes the more general kernel calibration error. Zhang et al. [2020] and Gupta et al. [2020] also consider kernel-based calibration. However, these methods only consider the similarity between model confidences $\hat{p}(x), \hat{p}(x')$, rather than the inputs $x, x'$ themselves.

## 3 THE LOCAL CALIBRATION ERROR

Recall that commonly used metrics for calibration, such as the ECE or the MCE, are global in nature and thus only measure an *aggregate* reliability over the entire dataset, making them insufficient for many applications. An ideal calibration metric would instead measure calibration at an individual level; however, doing so is impossible without making assumptions about the ground truth distribution [Zhao et al., 2020]. A localized calibration metric represents an adjustable balance between these two extremes. Ideally, such a metric should measure calibration at a local level (where the extent of the local neighborhood can be chosen by the user) and group similar data points together.

In this section, we introduce the local calibration error (LCE), a kernel-based metric that allows us to measure the calibration locally around a prediction. Our metric leverages learned features to automatically group similar samples into a soft neighborhood, and allows the neighborhood size to be set with a hyperparameter $\gamma$. We also consider only points with a similar model confidence as the prediction, so that similarity is defined in terms of distance both in the feature space and in model confidence. Thus, the LCE effectively creates soft groupings that depend on the feature space; with a semantically meaningful feature space, these groupings correspond to useful subsets of the data. We then mention a few design choices and visualize LCE maps over a 2D feature space to show that we can use our metric to diagnose regions of local miscalibration.

## 3.1 LOCAL CALIBRATION ERROR METRIC

We propose a metric to measure calibration locally around a given prediction. The calibration of similar samples should be similar, so we use a kernel similarity function $k_\gamma : \mathcal{X} \times \mathcal{X} \to \mathbb{R}_+$, which provides similarity scores, to define soft local neighborhoods. $k_\gamma(x, x')$ has bandwidth $\gamma > 0$, which determines the extent of the local neighborhood — as $\gamma$ increases, the neighborhood grows. Less similar (i.e., further away) samples $x'$ have less influence on the local calibration metric at $x$. Also, as with the ECE and MCE (Eqs. 2 and 3), we use binning and consider only the points in the same confidence bin as $x$. Thus, the samples that influence the local calibration metric at $x$ are similar to $x$ in both features and model confidences.

More formally, let $\phi : \mathcal{X} \to \mathbb{R}^d$ be a feature map that transforms an input to a feature vector, and let $k_\gamma$ be parameterized as $k_\gamma(x, x') = g((\phi(x) - \phi(x'))/\gamma)$ for some Lipschitz function $g : \mathbb{R}^d \to \mathbb{R}_+$. Then given a data point $x \in \mathcal{X}$ and a classifier $(f, \hat{p})$, the *local calibration error* (LCE) of the model at $x$ is the expected difference between the model's confidence and accuracy on a randomly sampled data point $x' \sim \text{Pr}$, weighted by the kernel similarity $k_\gamma(x, x')$.

We say a probabilistic classifier $(\hat{p}, f)$ is *perfectly locally calibrated* with respect to $k_\gamma$ if

$$\sup_{x \in \text{supp}(\text{Pr})} \underbrace{\left( \begin{array}{c} \mathbb{E}_{(x', y') \sim \text{Pr}} \Big[ (\hat{p}(x') - \mathbb{1}[f(x') = y']) \\ \cdot k_\gamma(x, x') \,\Big|\, \hat{p}(x') = \hat{p}(x) \Big] \end{array} \right)}_{:=\text{LCE}_\gamma^\star(x; f, \hat{p})} = 0.$$

Similar to perfect calibration, perfect local calibration is achieved by the Bayes-optimal classifier. In general, perfect local calibration is a much stricter notion than perfect calibration due to localizing to each indiviudal data point $x$, and reduces to perfect calibration if $k_\gamma(x, x') \equiv 1$ is a trivial kernel.

To define LCE on a finite dataset, we perform an additional binning on the confidence to deal with the conditioning.

Let $\mathcal{D} = ((x_1, y_1), \ldots, (x_N, y_N))$ be a dataset, and let $\beta(x) = \{ i : \hat{p}(x_i) \in B(\hat{p}(x)) \}$ be the set of indices of the points in $\mathcal{D}$ occupying the same confidence bin as $x$. Then we can compute the LCE by

$$\text{LCE}_\gamma(x; f, \hat{p}) = \left| \frac{\sum_{i \in \beta(x)} (\hat{p}(x_i) - \mathbb{1}[f(x_i) = y_i]) k_\gamma(x, x_i)}{\sum_{i \in \beta(x)} k_\gamma(x, x_i)} \right|. \quad (4)$$

Note that the quantity $(\hat{p}(x_i) - \mathbb{1}[f(x_i) = y_i])$ is simply the difference between the confidence and the accuracy for sample $x_i$, and the denominator is a normalization term.

We then define the maximum local calibration error (MLCE) as

$$\text{MLCE}_\gamma(f, \hat{p}) := \max_x \text{LCE}_\gamma(x; f, \hat{p}). \qquad (5)$$

Intuitively, the LCE considers a neighborhood about a sample $x$ (as defined by the kernel $k_\gamma$ and the confidence bin $B$), and computes the kernel-weighted average of the difference between the confidence and accuracy for each sample in that neighborhood. Note that by changing the bandwidth $\gamma$, we can interpolate the LCE between an individualized calibration metric (as $\gamma \to 0$) and a global one (as $\gamma \to \infty$). Lemma 1 makes this more concrete under the assumption that $\lim_{\gamma \to \infty} k_\gamma(x, x') = 1$ (proof in Appendix C). For example, the Laplacian and Gaussian kernels satisfy this condition.

**Lemma 1.** *As $\gamma \to \infty$, the MLCE converges to the MCE.*

Theorem 1 shows that under certain regularity conditions, the finite-sample estimator $\text{LCE}(x)$ converges uniformly and sample-efficiently to its true expected value $\text{LCE}^*(x)$:

**Theorem 1.** *(Informal) Let* $\alpha \leq \inf_{x \in \mathcal{X}} \mathbb{E}\left[k_\gamma(X, x)\mathbb{1}[\hat{p}(X) \in B(\hat{p}(x))]\right]$ *be a lower bound on the expectation of the kernel, and $d$ be the dimension of the kernel's feature space. If the sample size is at least $\widetilde{O}(d/\alpha^4\epsilon^2)$ where $\epsilon > 0$ is a target accuracy level, then with probability at least $1 - \delta$ we have*

$$\sup_{x \in \mathcal{X}} \left| \text{LCE}_\gamma(x; f, \hat{p}) - \text{LCE}^*_\gamma(x; f, \hat{p}) \right| \leq \epsilon.$$

*Here, $\widetilde{O}$ hides log factors of the form $\log(1/\alpha\gamma\delta\epsilon)$. In practice, $\alpha$ depends inversely on $\gamma$.*

To summarize, the MLCE measures a worst-case individual calibration error as $\gamma \to 0$ (i.e., the effective neighborhood is very small) and converges to the global MCE metric as $\gamma \to \infty$ (i.e., the effective neighborhood is very large). In practice, one must pick intermediate values of $\gamma$ to balance a more local notion of calibration error with the sample efficiency of its estimation. A more formal statement and full proof of Theorem 1 can be found in Appendix D.

## 3.2 CHOICE OF KERNEL AND FEATURE MAP

In this work, we compute the LCE using 15 equal-width bins and use the Laplacian kernel

$$k_\gamma(x, x') = \exp\left(-\frac{\|\phi(x) - \phi(x')\|_1}{d\gamma}\right).$$

Because distances in a high-dimensional input space (e.g., image data) may not be meaningful on their own, we evaluate the kernel on a feature representation of $x$ rather than on $x$ itself. Features learned from neural networks have

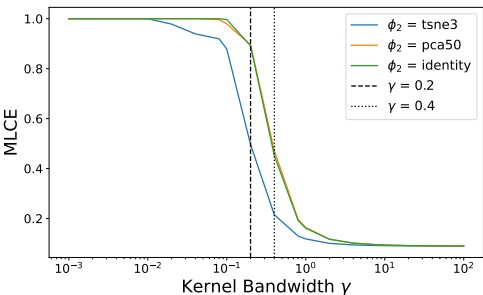

Figure 1: MLCE of a Resnet-50 classifier on the ImageNet test split, as a function of the kernel bandwidth $\gamma$. We use a Laplacian kernel with feature map $\phi_2 \circ \phi_1$, where $\phi_1 : \mathcal{X} \to \mathbb{R}^{2048}$ is the Inception-v3 model's hidden layer. Blue: $\phi_2 : \mathbb{R}^{2048} \to \mathbb{R}^3$ is t-SNE; orange: $\phi_2 : \mathbb{R}^{2048} \to \mathbb{R}^{50}$ is PCA; green: $\phi_2(z) = z$ is the identity.

proven useful for a wide range of tasks, and they have been shown to capture useful semantic features of their inputs [Huh et al., 2016, Chen et al., 2020, Li et al., 2020]. The kernel similarity term $k_\gamma(x, x')$ in the LCE thus leverages learned features to *automatically* capture rich subgroups of the data. For image data, we chose an Inception-v3 model as our feature map, since Inception features are widely accepted as useful and representative in many areas (e.g., for generative models [Salimans et al., 2016]), though other neural features can also be used (Appendix B). For tabular data, we used the final hidden layer of the neural network trained for classification.

In general, we also use t-SNE or PCA to reduce the dimension of the feature space. For example, the 2048-D Inception-v3 embedding is still very high-dimensional. We report results using t-SNE to reduce the dimension to 2 or 3, as well as PCA to reduce the dimension to reduce the dimension to 50 for image data and 20 for tabular data. Thus the overall representation function is $\phi(x) = \phi_2(\phi_1(x))$, where $\phi_1$ maps from the inputs to the neural features, and $\phi_2$ reduces the feature space dimension.

Figure 1 plots the MLCE as a function of the kernel bandwidth for an ImageNet classification task. Note that when $\gamma$ is small, the MLCE is 1 (a worst-case individual calibration error), and when $\gamma$ is large, the MLCE approaches the global MCE. To obtain a single summary statistic describing the local calibration error, we can view this plot and pick a value of $\gamma$ between the limiting behaviors. We find that $\gamma = 0.2$ and $\gamma = 0.4$ are good intermediate points for the 3-D t-SNE and 50-D PCA features, respectively (Figure 1).

## 3.3 LOCAL CALIBRATION ERROR VISUALIZATIONS

To provide more intuition for the LCE, we will now visualize some examples of the LCE metric over a 2-D

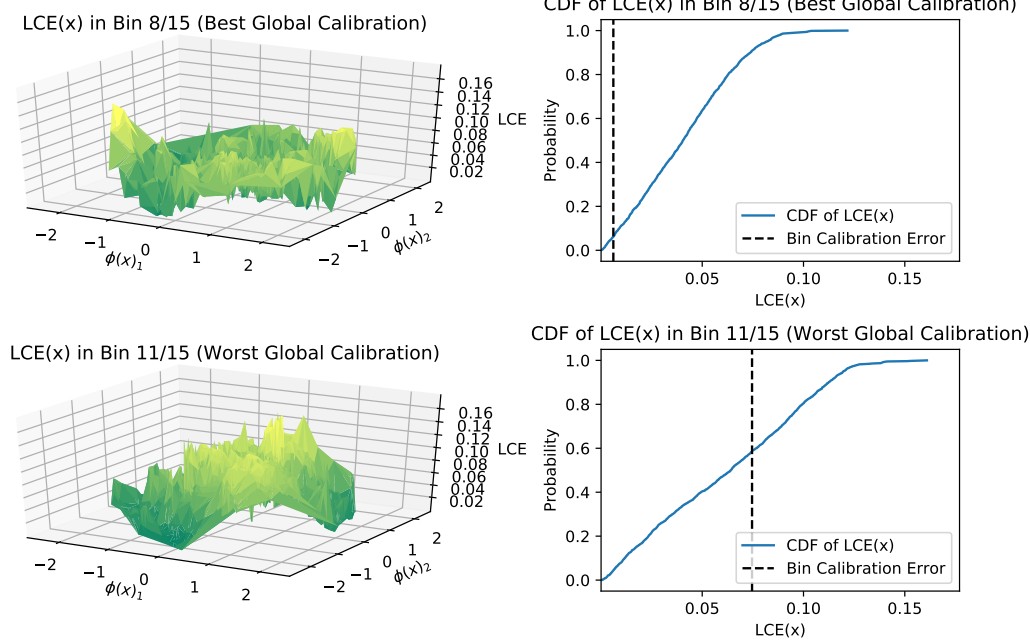

Figure 2: We visualize $\mathrm{LCE}_{0.2}(x; f, \hat{p})$ for a ResNet-50 classifier $(f, \hat{p})$ pre-trained on ImageNet, for every image $x$ in the ImageNet validation set. We focus on the bins with the best and worst global calibration errors.

feature embedding. We consider a ResNet-50 model pre-trained on ImageNet as our classifier $(f, \hat{p})$, and pre-trained Inception-v3 features as a feature map $\phi_1 : \mathcal{X} \to \mathbb{R}^{2048}$. $\phi_2 : \mathbb{R}^{2048} \to \mathbb{R}^2$ then reduces the 2048-D feature vectors with t-SNE to two dimensions for ease of visualization in the LCE landscapes, so our overall representation function is $\phi = \phi_2(\phi_1(x))$. Figure 2 visualizes the landscape of $\mathrm{LCE}_{0.2}(x; f, \hat{p})$ as a function of $\phi(x)$ for the entire ImageNet validation set, as well as the marginal CDF of $\mathrm{LCE}_{0.2}(x, f, \hat{p})$. We show these visualizations for the two confidence bins with the best and worst *global* calibration.

In the bin with the best global calibration, Figure 2 (top) shows that the landscape of the LCE is highly non-uniform, and the CDF of the LCE lies almost entirely to the right of the bin's average calibration error. Numerically, the bin's average calibration error is $0.0061$, while its average LCE is $0.0411$. This implies that the regions where the model is underconfident and overconfident are spatially clustered within the bin. Because global calibration metrics solely consider the average accuracy and average confidence within a bin, confidence predictions that are too high and too low are averaged out to obtain a low overall error value; they fail to capture this *localized* miscalibration.

In the bin with the worst global calibration, Figure 2 (bottom) clearly shows that the LCE still has high variance, even though the average calibration error of the bin ($0.0746$) is much closer to its average LCE ($0.0629$). The CDF plot provides more evidence that the landscape is not flat — there is no sharp rise at the bin calibration error. However, in this

case the regions that are underconfident and overconfident are not clustered spatially.

## 4 LCE RECALIBRATION

In this section, we introduce **lo**cal **re**calibration (LoRe), a non-parametric recalibration method that adjusts a model's output confidences to achieve better local calibration. Our method improves the LCE more than existing recalibration methods, and using our method improves performance on both downstream fairness tasks and downstream decision-making tasks. Specifically, we can leverage the kernel similarity to achieve strong calibration for all sensitive subgroups of a population, without knowing those groups a priori. As long as the feature space is semantically meaningful, LoRe provides utility for downstream tasks without needing subgroup labels for the samples. If the subgroups *are* known, one can recover standard group-wise recalibration methods (and metrics) by using the improper kernel $k(x, x') = \mathbb{1}[x, x' \text{ in same group}]$.

The idea behind our method is simple: we can compute the kernel-weighted accuracy for each point $x$ of all the points that are in the same confidence bin as $x$, and then reset the confidence of $x$ to this kernel-weighted accuracy value. Note that using the kernel function to compute this value is intuitively like taking a weighted average of the accuracy of the points in the local neighborhood of $x$. Thus, LoRe can be considered a local analogue to histogram binning.

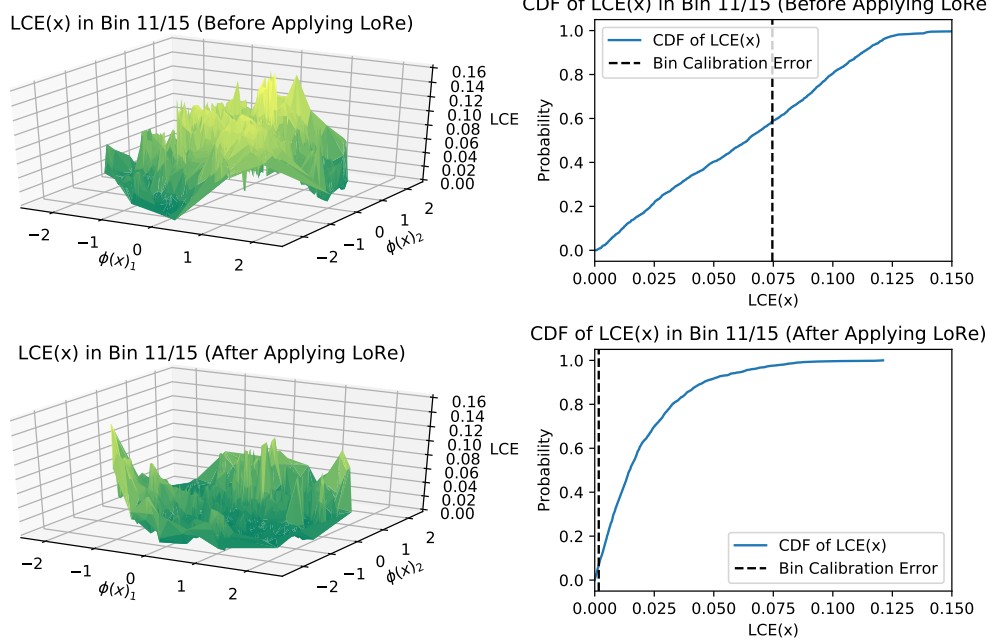

Figure 3: We visualize $\text{LCE}_{0.2}(x; f, \hat{p})$ for a ResNet-50 classifier $(f, \hat{p})$ pre-trained on ImageNet, for every image $x$ in the ImageNet validation set for bin 11 (the bin with the worst global calibration before applying LoRe).

More formally, given a trained classifier $(f, \hat{p})$, a recalibration dataset $\mathcal{D} = ((x_1, y_1), \ldots, (x_N, y_N))$, and a fixed point $x \in \mathcal{X}$, let $\beta(x) = \{i : \hat{p}(x_i) \in B(\hat{p}(x))\}$ be the set of indices of the points in $\mathcal{D}$ occupying the same confidence bin as $x$. Then, we compute the recalibrated confidence as

$$\hat{p}'(x) = \frac{\sum_{i \in \beta(x)} k_\gamma(x, x_i) \mathbb{1}[f(x_i) = y_i]}{\sum_{i \in \beta(x)} k_\gamma(x, x_i)}. \qquad (6)$$

Equation 6 represents the kernel-weighted average accuracy of all points in the same confidence bin as $x$. In the limit as the kernel bandwidth $\gamma \to \infty$, $\hat{p}'(x) \to \sum_{i \in \beta(x)} \mathbb{1}[f(x_i) = y_i]/|\beta(x)|$ recovers histogram binning. As $\gamma \to 0$, $\hat{p}'(x) \to \mathbb{1}[f(x_{i*}) = y_{i*}]$, where $i^* = \arg\min_{i \in \beta(x)} k_\gamma(x, x_i)$, thus recovering a nearest-neighbor method. For intermediate $\gamma$, our method interpolates between the two extremes. Throughout this work, we used $\gamma = 0.2$ for LoRe with tSNE and $\gamma = 0.4$ for LoRe with PCA throughout this work, since these represent intermediate points between the limiting behaviors of the LCE (e.g., see Fig. 1).

In Figure 3, we visualize the landscape of $\text{LCE}_{0.2}(x; f, \hat{p})$ for the ImageNet validation set both before and after applying LoRe, for bin 11 (the bin with the worst global calibration before applying LoRe). Note that the LCE landscape before applying LoRe has an area that is raised relative to the rest of the landscape, indicating systematic miscalibration. However, after applying LoRe, the landscape is both flatter and lower, indicating improved global and local calibration, and indeed the CDF plot shows a sharper rise than before. Thus, LoRe works as desired.

## 5 EXPERIMENTS

In this section, we show empirically that LoRe substantially improves LCE values, and that these lower LCE values lead to better performance on downstream fairness and decision-making tasks. In particular, we evaluate the local calibration through the MLCE, because we are interested in understanding a model's *worst-case* local miscalibration. On each task, we compare the performance of LoRe to no recalibration ('Original'), temperature scaling ('TS') [Guo et al., 2017], histogram binning ('HB') [Zadrozny and Elkan, 2001], isotonic regression ('IR') [Zadrozny and Elkan, 2002], and direct MMCE optimization ('MMCE') [Kumar et al., 2018], all strong *global* recalibration methods.

We first run extensive experiments on four datasets to demonstrate that LoRe outperforms all baselines and achieves the lowest MLCE over a wide range of $\gamma$ values. We then evaluate the performance of our method on a fairness task, where it is important that a model is well-calibrated for all sensitive subgroups of a given population, and we demonstrate that it achieves the lowest group-wise MCE. Notably, we find that the MLCE is well-correlated with the group-wise MCE across all experimental settings, and thus achieving low MLCE is a good indicator that a model has good group-wise calibration. Finally, we compare our method against the baselines on a cost-sensitive decision-making task, where there is a low cost for a prediction of "unsure" but a high cost for an incorrect prediction, and show that our method achieves the lowest cost.

| Recalibration method | Setting 1 | Setting 2 | Setting 3 | Setting 4 |
|---|---|---|---|---|
| No recalibration | 0.588 ± 0.107 | 0.407 ± 0.087 | 0.446 ± 0.083 | 0.480 ± 0.122 |
| Temperature scaling | 0.521 ± 0.092 | 0.532 ± 0.089 | 0.441 ± 0.079 | 0.403 ± 0.108 |
| Histogram binning | 0.515 ± 0.081 | 0.218 ± 0.056 | 0.268 ± 0.067 | 0.368 ± 0.108 |
| Isotonic regression | 0.596 ± 0.063 | 0.615 ± 0.100 | 0.716 ± 0.082 | 0.425 ± 0.047 |
| MMCE optimization | 0.526 ± 0.172 | 0.429 ± 0.079 | 0.475 ± 0.079 | 0.411 ± 0.088 |
| Group temp. scaling | 0.423 ± 0.066 | 0.673 ± 0.075 | 0.329 ± 0.108 | 0.411 ± 0.110 |
| Group hist. binning | 0.542 ± 0.083 | 0.260 ± 0.053 | 0.352 ± 0.068 | 0.414 ± 0.090 |
| LoRe (tSNE) (ours) | **0.351 ± 0.084** | **0.165 ± 0.055** | 0.235 ± 0.063 | **0.215 ± 0.037** |
| LoRe (PCA) (ours) | 0.392 ± 0.071 | 0.167 ± 0.013 | **0.154 ± 0.082** | 0.300 ± 0.065 |

Table 1: Performance on downstream fairness, as measured by maximum group-wise MCE (lower is better). Experimental settings as described in Section 5.3. Mean and standard deviations are computed over 60 random seeds for settings 1 and 4, and 20 for settings 2 and 3. Best results are **bold**.

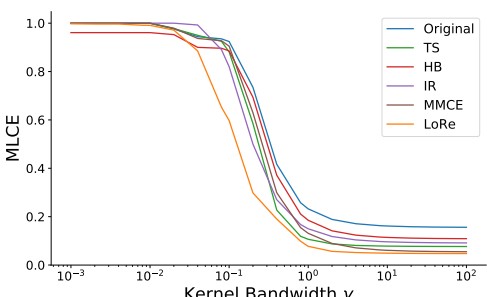

Figure 4: MLCE vs. kernel bandwidth $\gamma$ for ImageNet. LoRe (with t-SNE and $\gamma = 0.2$) achieves the lowest MLCE for a wide range of $\gamma$. This suggests that LoRe leads to lower LCE values across the whole dataset.

| | Setting 1 | Setting 2 | Setting 3 | Setting 4 |
|---|---|---|---|---|
| ECE | 0.102 | -0.061 | -0.195 | 0.012 |
| MCE | 0.233 | 0.439 | 0.281 | 0.387 |
| NLL | 0.542 | 0.045 | -0.287 | 0.051 |
| Brier | 0.101 | 0.144 | -0.280 | 0.024 |
| MLCE (tSNE) | **0.642** | **0.801** | 0.591 | 0.566 |
| MLCE (PCA) | 0.639 | 0.659 | **0.778** | **0.603** |

Table 2: Pearson correlation between max group-wise MCE and other calibration metrics (higher is better). Experimental settings as described in Section 5.3. Best results in **bold**. We use $\gamma = 0.2$ for MLCE (tSNE) and $\gamma = 0.4$ for MLCE (PCA). MLCE is better-correlated with the max group-wise MCE than any of the global metrics in all settings.

## 5.1 DATASETS

**ImageNet dataset [Deng et al., 2009]:** A large-scale dataset of natural scene images with 1000 classes; over 1.3 million images total. The training/validation/test split is 1.3mil / 25,000 / 25,000.

**UCI Communities and Crime dataset [Dua and Graff, 2017]:** This tabular dataset contains attributes about American neighborhoods (e.g., race, age, employment, housing, etc.). The task is to predict the neighborhood's violent crime rate. The training/validation/test split is 1494 / 500 / 500. We randomize this split over multiple trials.

**CelebA dataset [Liu et al., 2015]:** A large-scale dataset of face images with 40 attribute annotations (e.g., glasses, hair color, etc.); 202,599 images total. The training/validation/test split is 162,770 / 19,867 / 19,962.

**COMPAS Criminal Recidivism dataset [Dieterich et al., 2016]:** This tabular dataset reports American individuals' demographic information and criminal history. The task is to predict whether a given offender will commit another violent crime within 2 years. The training/validation/test split is 2,020 / 1,000 / 1,000. We randomize this split over multiple trials.

## 5.2 RECALIBRATION PERFORMANCE

LoRe substantially improves the LCE values. In Figure 4, we plot the MLCE as a function of $\gamma$. We can see that our method outperforms all baselines (strong global calibration methods) across a wide range of $\gamma$ values on ImageNet. This is true despite the fact that we only implement LoRe for a single $\gamma$. Appendix B provides similar results on the Communities & Crime, CelebA, CIFAR-10, and CIFAR-100 datasets. Note that LoRe works well regardless of the feature map and the dimensionality reduction method (see Section 5.3 for results with both t-SNE and PCA). Although the results shown in this section use Inception-v3 features, we show similar results in Appendix B with AlexNet [Krizhevsky, 2014], DenseNet121 [Huang et al., 2018], and ResNet101 [He et al., 2015] features.

Recall that as $\gamma$ gets large, the MLCE recovers the MCE; because LoRe does well even at large $\gamma$, our method also works well at minimizing global calibration errors. The fact that LoRe lowers the worst-case LCE suggests that it leads to lower LCE values across the entire dataset.

| Recalibration method | Setting 1 | | | Setting 2 | | | Setting 3 | | | Setting 4 | | |
|---|---|---|---|---|---|---|---|---|---|---|---|---|
| | ECE(%) | NLL | Brier | ECE(%) | NLL | Brier | ECE(%) | NLL | Brier | ECE(%) | NLL | Brier |
| No recalibration | $15.1_{2.7}$ | $.96_{25}$ | $.17_{02}$ | $\mathbf{1.8_{0.3}}$ | $\mathbf{.617_{004}}$ | $.641_{004}$ | $1.1_{0.3}$ | $.782_{006}$ | $.571_{006}$ | $3.6_{1.4}$ | $.408_{023}$ | $\mathbf{.124_{008}}$ |
| Temperature scaling | $4.9_{1.7}$ | $.43_{03}$ | $.14_{01}$ | $2.0_{0.3}$ | $.619_{004}$ | $.622_{003}$ | $\mathbf{1.0_{0.2}}$ | $\mathbf{.781_{006}}$ | $.569_{002}$ | $2.9_{1.2}$ | $\mathbf{.405_{021}}$ | $\mathbf{.124_{007}}$ |
| Histogram binning | $\mathbf{3.3_{1.1}}$ | $.48_{03}$ | $.15_{0.1}$ | $2.5_{0.2}$ | $.619_{004}$ | $.614_{003}$ | $2.5_{0.4}$ | $.788_{006}$ | $.552_{002}$ | $2.7_{1.1}$ | $.414_{023}$ | $.126_{008}$ |
| Isotonic regression | $30.6_{2.3}$ | $.79_{05}$ | $.30_{02}$ | $2.6_{0.2}$ | $.618_{004}$ | $.615_{003}$ | $2.4_{0.2}$ | $.785_{006}$ | $.553_{002}$ | $34.4_{1.7}$ | $.707_{021}$ | $.253_{004}$ |
| MMCE optimization | $4.4_{1.3}$ | $.43_{03}$ | $.14_{01}$ | $3.8_{0.7}$ | $.646_{014}$ | $.679_{012}$ | $5.4_{0.8}$ | $.808_{009}$ | $.619_{009}$ | $3.4_{1.9}$ | $.415_{039}$ | $.125_{009}$ |
| LoRe (tSNE) (ours) | $3.5_{1.1}$ | $\mathbf{.42_{02}}$ | $\mathbf{.13_{01}}$ | $2.8_{0.2}$ | $.623_{004}$ | $.613_{003}$ | $2.6_{0.3}$ | $.792_{006}$ | $.551_{002}$ | $2.7_{1.1}$ | $.410_{022}$ | $.125_{007}$ |
| LoRe (PCA) (ours) | $4.5_{1.4}$ | $.44_{02}$ | $.14_{01}$ | $3.1_{0.2}$ | $.628_{004}$ | $\mathbf{.606_{003}}$ | $2.8_{0.4}$ | $.792_{007}$ | $\mathbf{.538_{002}}$ | $\mathbf{2.5_{1.0}}$ | $.410_{021}$ | $.126_{008}$ |

Table 3: Performance on global calibration metrics, formatted as mean$_{sd}$. Lower is better. Experimental settings as described in Section 5.3. Best results are **bold**. Across all settings, LoRe generally achieves a global calibration error that is comparable to the baselines.

## 5.3 DOWNSTREAM FAIRNESS PERFORMANCE

**Experimental Setup** In many fairness-related applications, it is important to show that a model is well-calibrated for all sensitive subgroups of a given population. For example, when predicting the crime rate of a neighborhood, a model should not be considered well-calibrated if it consistently underestimates the crime rate for neighborhoods of one demographic, while overestimating the crime rate for neighborhoods of a different demographic. Therefore, in this section, we examine the worst-case group-wise miscalibration of a classifier, as measured by the maximum group-wise MCE when evaluated only on sensitive sub-groups. We consider the following experimental settings:

1. UCI Communities and Crime: Predict whether a neighborhood's crime rate is higher than the median; group neighborhoods by their plurality race (White, Black, Asian, Indian, Hispanic). 60 random seeds for model training.
2. CelebA: Predict a person's hair color (bald, black, blond, brown, gray, other); group people by hair type (bald, receding hairline, bangs, straight, wavy, other). 20 random seeds for model training.
3. CelebA: Predict a person's hair type; group people by their hair color; inverse of Setting 2. 20 random seeds for model training.
4. COMPAS Criminal Recidivism: Predict whether an individual commits another violent crime within 2 years; group individuals based on race. 60 random seeds for model training.

For each task, we train a classifier (see Appendix A for full details) and recalibrate its output confidences using each of the recalibration methods.

**Results** Table 1 reports the maximum group-wise MCE for each of the recalibration methods on each of the three tasks. LoRe outperforms the other baselines, achieving an average 50% reduction over no recalibration and an average 24% improvement over the next best global recalibration method. (Figures 1, 2, 3, and 4 in Appendix B show that LoRe is also the best method of lowering the MLCE over a wide range of $\gamma$). Notably, LoRe is robust to the feature map used (tSNE vs. PCA). It even outperforms global methods applied to each individual group, implying that correcting local calibration errors is a robust way to improve group calibration that generalizes better than naive alternatives.

Moreover, Table 2 shows that the maximum group-wise MCE is well-correlated with the MLCE, and it is in fact *much* better correlated with MLCE than global calibration metrics. Taken together, our results indicate that lowering the LCE has positive implications in fairness settings that cannot be achieved by simply lowering global metrics like the ECE. For reference, we also include the performance of all methods on various global calibration metrics in Table 3, which shows that LoRe is able to improve worst-case group-wise calibration without meaningfully sacrificing (and in some cases improving) average-case global calibration.

## 5.4 DOWNSTREAM DECISION-MAKING

**Experimental Setup** Machine learning predictions are often used to make decisions, and in many situations an agent must select a best action in expectation. As an example, suppose there is a low cost $u$ associated with returning "unsure" and a high cost $w$ associated with returning an incorrect classification (e.g., in situations such as autonomous driving, being unsure incurs only the small cost of calling a human operator, but making an incorrect classification incurs a high cost). An agent with good uncertainty quantification can make a more optimal decision about whether to return a classification or return "unsure"; for a calibrated model, it would be optimal for the agent to return "unsure" below the confidence threshold of $1 - u/w$, and return a prediction above this threshold.

Following this policy — i.e., returning "unsure" when the confidence is below this threshold and returning a prediction when the confidence is above it, we used a ResNet-50 model to make predictions on ImageNet, and recalibrated the predictions with each of the recalibration methods. For each method, we then calculated the total reward attained under various reward ratios $w/u$, as well as various global calibration metrics.

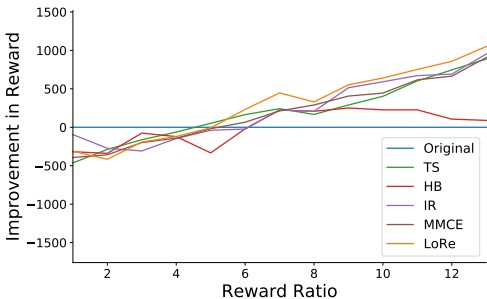

Figure 5: Reward attained vs. reward ratio for the ImageNet dataset (higher is better). LoRe achieves the highest rewards across a wide range of reward ratios.

| Recalibration method | ECE | NLL | Brier |
|---|---|---|---|
| No recalibration | 0.037 | 0.959 | 40.64 |
| Temperature scaling | 0.022 | 0.948 | 40.60 |
| Histogram binning | 0.012 | 0.952 | 40.59 |
| Isotonic regression | 0.011 | **0.945** | 40.59 |
| MMCE optimization | 0.061 | 0.965 | 40.67 |
| LoRe (ours) | **0.007** | 0.955 | **40.58** |

Table 4: Performance on global calibration metrics on ImageNet. Lower is better. Best results are **bold**. LoRe achieves strong global calibration according to all metrics.

**Results** In Figure 5, we show the improvement in the total reward over the original classifier (i.e., no recalibration) as a function of the reward ratio $w/u$ (the ratio of the cost of an incorrect classification to the cost of being unsure). Across a wide range of reward ratios, LoRe achieves the highest reward. The MLCE curves for this task are shown in Figure 4; note that LoRe also achieves lower LCE values than the global recalibration methods. Table 4 reports several global calibration metrics; LoRe achieves strong global calibration. These results indicate that our recalibration method most effectively lowers LCE values without sacrificing (and indeed often improving) average-case global calibration, and that these lower LCE values correspond to better performance on this decision-making task.

## 6 CONCLUSION

In this paper, we introduce the local calibration error (LCE), a metric that measures calibration in a localized neighborhood around a prediction. The LCE spans the gap between fully global and fully individualized calibration error, with an effective neighborhood size that can be set with a bandwidth parameter $\gamma$. We also introduce LoRe, a recalibration method that greatly improves the local calibration. Finally, we demonstrate that achieving lower LCE values leads to better performance on downstream fairness and decision-making tasks. In future work, we hope to further explore

alternative feature spaces to define similarity, since the quality of our metric depends on the quality of the feature space underpinning the notion of locality.

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
