# OpenReview forum: "Local Calibration: Metrics and Recalibration"
_auai.org/UAI/2022/Conference — UAI 2022 Poster_

### Official Review · Reviewer_rAhv · 2022-04-01

**Q2(1) Originality/Novelty:** 3
**Q2(2) Significance/Impact:** 3
**Q2(3) Correctness/Technical Quality:** 3
**Q2(6) Clarity Of Writing:** 3
**Q6 Overall Score:** 7
**Q8 Confidence In Your Score:** 3

**Q1 Summary And Contributions:**

The paper proposes a metric for estimating the "local calibration error", which is an intermediate step between individual and global calibration. It then goes on to present a re-calibration method, "LoRe" which optimizes for this metric.
The efficacy of the proposed method is then demonstrated on several benchmarks.

**Q2 Assessment Of The Paper:**

More detailed information regarding each of these aspects is given below:

**Q2(4) Quality Of Experiments (Optional):**

3: Good: The experimental evaluation is adequate, and the results convincingly support the main claims.

**Q2(5) Reproducibility:**

2: Fair: Key resources (e.g., proofs, code, data) are unavailable but key details (e.g., proof sketches, experimental setup) are sufficiently well-described for an expert to confidently reproduce the main results.

**Q3 Main Strengths:**

The paper tackles an important and interesting problem, the assessment of calibration for "individual" or "groups of" predictions.
It is generally well-written and clearly structured. The main contributions are technically novel and can provide an improvement over existing methods. Necessary theoretical foundations are provided. The experimental evaluation is technically sound and answers most relevant questions.

**Q4 Main Weakness:**

The paper does not provide code for the reproducibility of results. In my opinion, this is important to allow assessment of the validity of results and soundness of experimentation protocol.

The proposed metric has a bandwidth parameter gamma which allows it to flexibly interpolate between individual and global calibration. Guidance with respect to how this parameter should be chosen is generally missing and might be important such that the metric is not tuned to show intended results

The paper does not provide any experimental evidence, that the proposed LCE criterion actually measures what it intends to measure.
This could be shown in a small-scale simulation study.

**Q5 Detailed Comments To The Authors:**

The related work states that multi-calibration lacks a tractable algorithm. How did the authors arrive at this conclusion? Algorithm 1 in the paper states one, or am I missing anything here?

It would be interesting to see loss profiles before and after calibration with the LoRe method, does it change the loss profiles accordingly (it should, I guess).

I am not sure I would agree with the statement in Table 3 that LoRe generally achieves comparable global calibration given the results.

In general, the experimental evaluation compares the calibration method "LoRe" but does not study the proposed metric in depth.
I think this would be relevant if the proposed metric should be used in practice.

**Q7 Justification For Your Score:**

The paper is interesting, well-written and tackles an important problem. Contributions are technically novel and experimental evaluation shows that the proposed methods work.
Code for reproducibility is missing, and I would find a simulation experiment assessing the metric interesting.

**Q9 Complying With Reviewing Instructions:**

1: Yes.

---

### Official Review · Reviewer_zrhF · 2022-04-10

**Q2(1) Originality/Novelty:** 3
**Q2(2) Significance/Impact:** 3
**Q2(3) Correctness/Technical Quality:** 3
**Q2(6) Clarity Of Writing:** 3
**Q6 Overall Score:** 6
**Q8 Confidence In Your Score:** 4

**Q1 Summary And Contributions:**

This paper proposed a novel local calibration estimation method that utilizes a kernel to estimate individual calibration errors.
Then the authors further proposed a local recalibration method to improve calibration performance.

**Q2 Assessment Of The Paper:**

More detailed information regarding each of these aspects is given below:

**Q2(4) Quality Of Experiments (Optional):**

3: Good: The experimental evaluation is adequate, and the results convincingly support the main claims.

**Q2(5) Reproducibility:**

3: Good: Key resources (e.g., proofs, code, data) are available and key details (e.g., proofs, experimental setup) are sufficiently well-described for competent researchers to confidently reproduce the main results.

**Q3 Main Strengths:**

1. This paper is well written and easy to follow
2. The proposed method is well motivated. Existing local calibration methods rely on group information or randomized predictor, and a local calibration without such constraints is worth exploring.
3. The proposed local recalibration method achieved promising performance concerning the proposed local calibration metric and
global calibration metric.


**Q4 Main Weakness:**

1. The proposed method is effectively a soft grouping method using a kernel. After grouping, it is the same as grouping-based methods.
Thus, the technical contribution is limited.
2. In experiments, only global calibration methods are considered.
It would be better if the grouping-based method or randomized method such as "Individual Calibration with Randomized Forecasting" is compared.

**Q5 Detailed Comments To The Authors:**

Comments on weakness 2:
Since the proposed local calibration method does not use any group information or a randomized predictor,
the performance may not be comparable.
However, it is desired to show why the proposed kernelized grouping is valid by comparing with them.

**Q7 Justification For Your Score:**

This paper proposed an effective local calibration estimation method that utilizes a kernel to estimate individual calibration errors. Please consider the previous questions in the rebuttal period.

**Q9 Complying With Reviewing Instructions:**

1: Yes.

---

### Official Review · Reviewer_jk5n · 2022-04-12

**Q2(1) Originality/Novelty:** 2
**Q2(2) Significance/Impact:** 3
**Q2(3) Correctness/Technical Quality:** 3
**Q2(6) Clarity Of Writing:** 3
**Q6 Overall Score:** 5
**Q8 Confidence In Your Score:** 3

**Q1 Summary And Contributions:**

The paper introduces a novel method to calibrate individual predictions (local calibration) of a a classifier using kernel methods.
The contribution are both at theoretical level and include experimental validation.

**Q2 Assessment Of The Paper:**

More detailed information regarding each of these aspects is given below:

**Q2(4) Quality Of Experiments (Optional):**

3: Good: The experimental evaluation is adequate, and the results convincingly support the main claims.

**Q2(5) Reproducibility:**

2: Fair: Key resources (e.g., proofs, code, data) are unavailable but key details (e.g., proof sketches, experimental setup) are sufficiently well-described for an expert to confidently reproduce the main results.

**Q3 Main Strengths:**

The paper is clear to follow, the idea is pretty straightforward and it has useful implications. The assumptions are clearly stated and the work is properly framed within the related work.


**Q4 Main Weakness:**

The empirical part might be improved. Only three datasets are used. Only deep neural networks are employed in the modelling while the method should be applicable also to more simple models. A more comprehensive approach should be preferred. See for instance the one in [Kull et al., 2019].

**Q5 Detailed Comments To The Authors:**

The paper is well written and it is easy to understand.
However, there are some small details for potential improvements. For instance, I do not understand why the authors emphasize the ability of LoRe in improving fairness empirically (line 5, column 1, page 2) when the considered datasets are just three. I would personally avoid such a claim when the focus of the paper is on lowering local calibration error.

Additionally, I would also present better what you mean by 60/20 random seeds in the empirical part. From the paper and the supplementary material, it is not clear whether you performed multiple random splits of the data (i.e. multiple train, validation and test sets) and you trained the same model over these different splits, or you simply changed the random seed of the learning algorithm.

A small typo in the line 10, column 1, page 6: “used used”.


**Q7 Justification For Your Score:**

I think the empirical part is somehow lacking some aspects that are important, such as testing multiple learning algorithms. However, the theoretical part is well written and the presented idea has some useful implications. Therefore I would be for a borderline accept.

**Q9 Complying With Reviewing Instructions:**

1: Yes.

---

### Official Review · Reviewer_imTW · 2022-04-17

**Q2(1) Originality/Novelty:** 3
**Q2(2) Significance/Impact:** 3
**Q2(3) Correctness/Technical Quality:** 3
**Q2(6) Clarity Of Writing:** 4
**Q6 Overall Score:** 7
**Q8 Confidence In Your Score:** 4

**Q1 Summary And Contributions:**

The paper considers calibration in the predicted outcome. In particular, the paper proposes an adjustable local calibration that finds a balance between the goals of achieving overall and individualized calibration. The LoRe method, as well as the empirical results, is also presented.

**Q2 Assessment Of The Paper:**

More detailed information regarding each of these aspects is given below:

**Q2(4) Quality Of Experiments (Optional):**

3: Good: The experimental evaluation is adequate, and the results convincingly support the main claims.

**Q2(5) Reproducibility:**

3: Good: Key resources (e.g., proofs, code, data) are available and key details (e.g., proofs, experimental setup) are sufficiently well-described for competent researchers to confidently reproduce the main results.

**Q3 Main Strengths:**

The paper is well-written and easy to follow. The motivation, problem setup, and results are clearly presented. The proposed adjustable calibration procedure strikes a good balance between the overall calibration (that is not fine-grained enough) and individualized calibration (that might not be realistic for certain scenarios). The experiments on various data sets demonstrate the efficacy of the proposed LoRe approach.

**Q4 Main Weakness:**

The only thing I find that can make the paper clearer is the definition of the fairness notion. From the topic of interest, I would assume that by "fairness" the paper is referring to the Predictive Rate Parity (also known as Calibrated, or Sufficiency, in the fairness literature) fairness. The proper calibration is exactly what PRP fairness is seeking. It would be better if the paper can explicitly specify the fairness notion of interest, so that there is no potential misunderstanding.

**Q5 Detailed Comments To The Authors:**

As stated in Q4, an explicit definition of the fairness notion of interest would be very helpful for readers to clearly understand the intended usage of the word "fairness" in the paper.

**Q7 Justification For Your Score:**

The motivation and problem setup are very clear. The adjustable balance between overall and individualized calibration is demonstrated, which has a practical impact. The proposed approach is evaluated on various data sets and shows strong performance. The contribution is solid and well presented.

**Q9 Complying With Reviewing Instructions:**

1: Yes.

---

### Decision · Program_Chairs · 2022-05-15

**Decision:**

Accept (Poster)

**Comment:**

Meta Review: This paper proposes a new calibration method that calibrates individual predictions using a kernel. Calibrating individual predictions is deemed a challenging task. The authors proposed a local calibration error (LCE) to measure the average reliability of similar predictions where the similarity is captured by a kernel function. It is further shown that LCE can be estimated effectively and can therefore instruct a local calibration.

All reviewers agreed that both the presented theoretical and empirical results are sound. The paper is also praised for its clarity and presentation.